# Optimising the secondary use of primary care prescribing data to improve quality of care: a qualitative analysis

Erica Barbazza [1], Robert A Verheij,[2] Lotte Ramerman,[2] Niek Klazinga,[1] Dionne Kringos[1]

¹Department of Public and Occupational Health, Amsterdam UMC Locatie AMC, Amsterdam, The Netherlands
²Learning Health Systems Research Programme, Netherlands Institute for Health Services Research, Utrecht, The Netherlands

**Correspondence to**
Erica Barbazza;
e.s.barbazza@amsterdamumc.nl

## ABSTRACT

**Objectives** To explore available data sources, secondary uses and key considerations for optimising the actionability of primary care prescribing data to improve quality of care in the Dutch context.

**Design** An exploratory qualitative study was undertaken based on semi-structured interviews. We anchored our investigation around three tracer prescription types: antibiotics; benzodiazepines and opioids. Descriptive and explanatory themes were derived from interview data using thematic analysis.

**Setting** Stakeholders were sampled from across the micro (clinical), meso (organisational) and macro (policy) contexts of the Dutch primary care system.

**Participants** The study involved 28 informants representing general practitioners (GPs), community pharmacists, regional chronic care networks (care groups), academia and research institutes, insurers, professional associations, electronic health record (EHR) vendors and national authorities.

**Results** In the Netherlands, three main sources of data for improving prescribing in primary care are in use: clinical data in the EHRs of GP practices; pharmacy data in community pharmacy databases and claims data of insurers. While the secondary use of pharmacy and claims data is well-established across levels, the use of these data together with EHR data is limited. Important differences in the types of prescribing information needed by micro-meso-macro context are found, though the extent to which current indicators address these varies by prescription type. Five main themes were identified as areas for optimising data use: (1) measuring what matters, (2) increasing data linkages, (3) improving data quality, (4) facilitating data sharing and (5) optimising fit for use analysis.

**Conclusions** To make primary care prescribing data useful for improving quality, consolidated patient-specific data on the indication for a prescription and dispensed medicine, over time, is needed. In the Netherlands, the selection of indicators requires further prioritisation to better signal the appropriateness and long-term use of prescription drugs. Prioritising data linkages is critical towards more actionable use.

## INTRODUCTION

Improving prescribing practices has received increasing policy attention globally. This

### STRENGTHS AND LIMITATIONS OF THIS STUDY

⇒ Semi-structured interviews elicited first-hand insights into the secondary use of primary care prescribing data, filling this knowledge gap in the published literature.

⇒ Stakeholder interviews spanned all levels of the Dutch healthcare system and engaged varied perspectives, including community pharmacists and general practice, offering diverse insights.

⇒ Three tracer prescription types were selected to anchor discussions with stakeholders and the findings may not capture the nuances of all prescriptions.

⇒ Our study is deliberately exploratory in nature, thus patterns and experiences by stakeholder types require testing with a larger sample, including patients, before they can be generalised.

prioritisation follows concerning trends, including rising levels of antimicrobial resistance,[1 2] an epidemic of opioid use[3–5] and the increasing misuse of benzodiazepines.[6–8] In the Dutch context—like other gate-keeping models of primary care—general practitioners (GPs) function as the first-line for patient management and entry-point to secondary healthcare services. In effect, GPs together with community-based pharmacists are central to services including the issuing and refilling of outpatient prescription medicines.[9] Measuring the performance of services provided by GPs and community pharmacists (both key primary care providers) is fundamental to improve quality.[10] Hence, the use of quality indicators, as a measurement tool to quantify quality, is of critical importance.[11–13]

In the Netherlands, the far-reaching digitalisation of patient data and physician prescribing has long been recognised as a powerful resource for improving quality.[14–16] All GP practices (approximately 5000) record data in electronic health records (EHRs) supplied by 10 main EHR vendor brands on the market.[16] Since 2014, primary care prescriptions are issued electronically for

BMJ

dispensing medicines at one of approximately 2000 community pharmacies across the country.[16] The resulting electronic primary care prescribing data has secondary uses that extend across the micro (clinical care), meso (organisations and networks) and macro (policy) context of the Dutch healthcare system.[17]

However, as health services research has increasingly called attention to, the availability of data alone does not guarantee its *use* for quality of care-related decision-making.[18 19] The information produced should also be actionable.[20] The movement towards learning healthcare systems further attests to the critical role of actionable data as an integral part of healthcare delivery processes.[21 22] In primary care, given the critical potential of prescription data to indicate, for instance, inappropriate prescriptions, overprescribing, addiction issues or antimicrobial resistance trends, it is essential to ensure healthcare systems are optimally using available prescription data for learning and decision-making purposes towards quality improvement in practice.

In the data-rich context of the Netherlands, activity around the use of healthcare data is high: survey data finds Dutch GPs regularly receive as many as 10 different feedback reports.[23] This volume of activity has called into question the extent to which performance indicators are actually used for improvement purposes. Research on the secondary uses of healthcare data has been conducted in the context of Dutch hospitals,[24] out-of-hours care[25] and integrated care networks.[26] In the absence of an overview of routine primary care prescribing data sources, what and how available data is used for learning and improvement purposes across the healthcare system is unclear.

In this study, we set out to investigate the current secondary uses of primary care prescribing data for improving quality of care through the first-hand insights of stakeholders across the Dutch healthcare system. We also aimed to distil their views on opportunities to improve the use of prescribing data for quality of care-related decision-making. Importantly, the optimisation of secondary uses of primary care prescribing data is an intermediary step to improving care. Direct uses of prescribing data for patient care, such as for education purposes and shared decision-making, is also a key aspect to improve prescribing,[27–29] however, these uses are outside the scope of this study. To anchor our investigation and generate concrete, practical examples of prescribing data uses, we focused on three commonly prescribed types of prescriptions: antibiotics; benzodiazepines and opioids. The prescriptions are each of significant societal and public health importance[30 31] and vary in their aetiological and therapeutic use (infection control, psychological disorders and pain management, respectively). In combination, the selected prescription types can offer insights into the use of primary care prescribing data as a whole.

With this aim and focus, the study is guided by the following three questions: what are the available sources and characteristics of primary care prescribing data? How is this data currently used for improving quality of care?

And, what are the key considerations for optimising the secondary uses of primary care prescribing data?

## METHODS

### Design

An exploratory qualitative study design was employed.[32] Reporting adheres to the Consolidated Criteria for Reporting Qualitative Research.[33] Semi-structured interviews with stakeholders ranging the clinical (micro), organisational (meso) and policy (macro) context of the Dutch healthcare system were conducted for rich individual exchanges and practical insights across the healthcare system.[34] The research team included experts on healthcare performance intelligence, primary care, health information systems and the Dutch context. The primary researcher and interviewer is an experienced qualitative researcher and doctoral student on the actionability of healthcare performance indicators.

To operationalise the construct of actionable indicators, we drew from an existing definition depicting actionability as the two related constructs of *fitness for purpose*—information serving an intended decision-making function—and *fitness for use*—the ability to get the right information, into the right hands at the right time.[20] To explore fitness for purpose, the definition's differentiation of types of uses of indicators across healthcare systems was applied. This depiction of actionable indicators, together with our three main research questions, served as the framework for our interview guide. Specifically, the themes explored with informants included: sources of primary care prescribing data; current uses of prescribing data (anchored in the selected prescription types); and perceived actionability constraints (online supplemental file 1).

### Sample and recruitment selection

We defined our target informants by Dutch stakeholders across the micro-meso-macro contexts of the healthcare system with first-hand use of primary care prescribing data for monitoring, assessing and/or improving quality. We identified more than 20 different stakeholders, ranging: government health agencies; associations, including patient and professional groups; regional care networks; health professionals; EHR suppliers; insurers and researchers (online supplemental file 2). An initial listing was prepared based on reviews of key literature[16 35 36] and the expertise of the study team. The list was validated with an existing Dutch network (Data Expert Community), with the representation of national stakeholders working in the field of healthcare data. Feedback from the network was solicited at an in-person meeting in November 2019 in Utrecht, the Netherlands.

We used multiple methods to reach prospective informants affiliated to the stakeholders identified. First, we reviewed the webpages of target stakeholders for contacts and membership lists. Second, the authorship of literature related to primary care and medicines in the Dutch

context (eg, scientific articles, reports, evaluations, fact-sheets, presentations) was extracted. Third, the expertise of the study team and advice of external experts were solicited, and a snowballing approach was applied. In a similar way, some prospective participants served as contact mediating informants, suggesting alternative colleagues best suited for participating. Informants were invited to participate in the study via email by the authors (EB, RAV, LR) and received a document detailing the background, aim, scope and research questions.

### Data collection

Interviews were conducted over a 4-month period (November 2019 to February 2020). Interviews ranged 30–60 min in length. They were conducted both in-person and at-distance by phone, based on the proximity and preference of informants. In instances where informants requested to extend an invitation to colleagues, these interviews were conducted jointly. We also accommodated requests to answer questions in writing. With the agreement of informants, interviews were recorded and transcribed verbatim. Regular meetings with the full study team were organised to discuss the process and recurrent themes. The interviews were considered complete when the range of informants represented stakeholders spanning the micro-meso-macro levels of the healthcare system.

### Data analysis

Thematic analysis was used to analyse interview data[37] in an Excel tool developed in the approach of Meyer and Avery.[38] The analysis process included familiarisation with the data, development of a coding framework, coding, mapping and interpretation of results. The coding framework was developed based on the items of the semi-structured interviews: purposes of use; actors; indicators; data sources; analysis; dissemination; barriers and opportunities for improvement (online supplemental file 2). Additional themes were generated through open (unrestricted) coding in an inductive approach. The initial coding and clustering of themes were conducted by the primary researcher. To ensure validity of the findings, the results were regularly reviewed by the full study team. In reporting on the results by research question, verbatim quotes were extracted from the transcripts.

### Patient and public involvement

The preliminary findings were shared at an international scientific conference in 2021. The interaction with participants provided a unique opportunity for critically reflecting on the findings.

### RESULTS

### Characteristics of informants

In total, 53 informants were contacted of which 28 were interviewed representing 26 different stakeholders. Ten prospective informants referred to an alternative contact

**Table 1** Summary of informant characteristics

| Characteristics | Total informants N=28 | |
| --- | --- | --- |
| | n* | % |
| Healthcare system level (context) | | |
| Micro (clinical) | 1 (4) | 4 |
| Meso (organisational) | 11 | 39 |
| Macro (policy) | 9 | 32 |
| Cross-cutting (research, EHR supplier) | 7 | 25 |
| Type of stakeholder | | |
| Association (patient, professional) | 8 | 29 |
| Care group (network) | 2 | 7 |
| Government health agency | 9 | 32 |
| Health professional | 1 (4) | 4 |
| EHR supplier | 4 | 14 |
| Insurer | 1 | 4 |
| Research | 3 | 11 |
| Gender | | |
| Female | 8 | 29 |
| Male | 20 | 71 |

*Numbers in parentheses indicate the total number of informants when individuals with multiple affiliations are accounted for.
EHR, electronic health record.

within their team or organisation. Non-participants were either unresponsive (n=12) or unavailable due to time constraints (n=3). In either instance (contact mediating informants or non-participants), no healthcare system level or type of stakeholder was overly non-responsive to participation. See online supplemental file 3 for a detailed breakdown.

Two interviews were conducted with two informants present. In two other instances, information was collected via email exchange only, at the preference of the informant. No repeat interviews were carried out. Some informants held multiple affiliations. Notably, three informants were both health professionals and affiliated to another stakeholder, as signalled by totals included in parentheses in table 1. For the purposes of reporting, only one primary affiliation has been used (table 1). See online supplemental file 3 also for a detailed overview of informant characteristics.

### Sources and characteristics of primary care prescribing data

Three main sources of primary care prescribing data for secondary uses towards improving quality are in use in the Netherlands: clinical data in the EHRs of GP practices and dispensing data related to prescriptions dispensed in community pharmacy databases and claims for prescriptions of insurers.

Datasets which can be combined and supplemented with other information are available, specifically: the

**Table 2** Primary care prescribing data landscape in the Netherlands according to informants

| Data source | Repository | Coverage | Nature of information | Advantages | Limitations |
|---|---|---|---|---|---|
| Clinical | EHRs | All GP practices | Prescription level data with patient ids including complete medical history, diagnosis, lab tests and prescribed medicines. | Includes indication for prescription. Possibility to link across databases using unique patient identifier. Possible to link with comorbidities. | Lacks data on prescriptions filled and dispensed by pharmacist. No central database. Varied recording of data across EHR suppliers. |
| Pharmacy dispensing data of community pharmacist | Foundation of Pharmaceutical Statistics | Across community pharmacies | Patient-level information on dispensed medicines in pharmacy system, medication including type, dosage, other medications. | Complete overview of dispensed medicines by community pharmacies. | Lacks data on diagnosis and lab results. Excludes: prescriptions issued but not retrieved; over-the-counter medicines; prescriptions issued and dispensed in hospitals. |
| Claims (pharmacy, services) | Drug Information Project (Dutch Health Care Institute) | Across community pharmacies | Information on prescription (eg, dosage, quantity dispensed), prescriber, dispensing pharmacy and price declared/ reimbursed filled by public pharmacies. | Data collected across all practices/public pharmacies. | Lacks data on diagnosis. Includes data only for reimbursed medicines and services. |
| Other repositories | Nivel Primary Care Database (Nivel) | Affiliate GP practices from across the country* | Data on consultations, diagnosis, prescribed medicines, with the possibility to link other data sources for environmental characteristics, migration background, income, insurance claims, pharmacy data. | Possibility to combine and supplement EHR data with information about pharmaceutical care and secondary level care. | EHR data from affiliated practices only, though representation across the country (10% of the population). |
| | Pharmo Data Network (Pharmo) | Affiliate care groups† | Linked data from public pharmacy database, GP database, hospital pharmacy databases, clinical laboratories. | Possibility to link to EHR data to administrative insurance claims data and pharmacy data. | Data from affiliate care groups only. |
| | Academic GP network databases | Networks in catchment area of large university hospitals | Patient-level data including complete medical history, diagnosis, medications, etc for affiliated practices. | Includes indication for prescription. Possibility to link across databases using unique patient identifier. | Limited to affiliate GP practices. Research-specific uses of data. |
| | Vektis database (Vektis) | Across health care insurers | Insurers claims database of all reimbursed services with data on physician services (eg, reason for visit) and procedures (eg, tests). | Completeness of database, with data spanning across the Dutch population and insurers. | Lacks data on diagnosis. Includes data only for reimbursed medicines and services. |

*Approximately 500 GP practices, 1.7 million patients.
†Approximately 13 care groups, 4 million patients.
EHR, electronic health record; GP, general practitioner; Nivel, Netherlands Institute for Health Services Research; Pharmo, Institute for Drug Outcomes Research Database.

Institute for Drug Outcomes Research Database,[39] Nivel Primary Care Database[15 35 40] and various research-specific datasets of academic networks of GPs (eg, Registration Network Groningen[41]). These datasets have the advantage of more complete information (diagnosis and dispensed medicines) though are limited to the voluntary participation GP practices. Other types of prescribing data though not specific to primary care include self-reported or physician-reported medicines' side effects[42] and inpatient prescribing in hospital databases.

Table 2 summarises these data sources, the nature of information and advantages, and limitations of each for secondary quality-related uses as described by informants. According to informants, not one data source is considered *complete*, as each has unique advantages, but also limitations as a potential source for quality-related decision-making. For example, clinical data in EHRs captures the diagnosis (indication) for a prescription, however, depending on the EHR system, it can lack details on the medicines retrieved and dispensed in community pharmacies. Conversely, administrative pharmacy data and insurance claims are rich in details of prescriptions dispensed and reimbursed, though lack clinical details found in EHRs, specifically associated laboratory results and a specific diagnosis. As informants described:

> The missed link between the diagnosis in the EHR and what is dispensed as the medication, leaves little

**Table 3** Examples of information needs by type of prescription as described by informants

| Context | Antibiotics | Benzodiazepines | Opioids |
|---|---|---|---|
| Macro (policy) | What is the overall volume of antibiotics prescribed annually? | How many elderly patients have a long-term benzodiazepine prescription? | What is the overall volume of opioids prescribed? How many are chronic opioid users? |
| Meso (organisational) | How does the volume of prescribing compare with previous years? (care groups) | How does the volume of prescribing compare with previous years and age groups? | How does the volume of prescribing compare with previous years and age groups? |
| Micro (clinical) | Have I prescribed antibiotics appropriately for infections? | How many of my patients have a long-term prescription? How many prescriptions were new vs refills? | How many of my patients have a long-term prescription? How many prescriptions were new vs refills? |

insights into whether the prescription provided was the right one or necessary. (Health professional–2)

From the pharmacist perspective, the absence of a link to a specific diagnosis means that interpreting values requires in most instances more analysis and reflection. (Association–13)

### Secondary uses of primary care prescribing data

The secondary uses and sources of primary care prescribing data are summarised to follow. See online supplemental file 1 for a detailed table. These descriptions are anchored in the illustrative prescription types applied. At the outset, the information needs by decision-making context and prescription type were described by informants (table 3).

### Micro level

Claims data of insurers is used to provide feedback on the quality of prescribing to GPs in a report called 'practice mirrors' introduced in 2018. These feedback reports detail the volume and costs of prescriptions and can signal GPs that overuse or underuse prescription medications. GPs participating to the Nivel, Pharmo or academic GP research network datasets receive additional feedback on their prescribing patterns.

Nearly all GPs in the Netherlands participate in pharmacotherapy audit groups (FTOs). FTOs are organised locally and are a practical mechanism for creating linkages between GPs and the pharmacists. As one informant described:

From my experience as a GP, the FTO is a great mechanism for linking up the GP and the pharmacists as the pharmacist really is the one that has a lot of data on what medicines are being handed out. The pharmacist has a really powerful dataset but they do miss the facts about the patient's actual needs. The linkage [exchange] between a GP and the pharmacists data set happens only at the meeting [FTO] itself. (Health Professional–2)

Informants described the indicators reported at the micro level vary for reasons primarily due to the type of data available to stakeholders, the priorities of practices and the relevance of existing indicators. On the latter,

informants noted differences between feedback that may be useful for a pharmacist versus a GP. For example, from the perspective of pharmacists, the following was described regarding benzodiazepines over an extended period of time:

There are some indicators to give feedback to pharmacists about whether they give long-term prescriptions to elderly people. But we do not use this as a quality indicator because the pharmacist's care is just a small amount of the care that is provided to patients using benzodiazepines … It depends [rather on] the work of the GPs. (Association–1)

In contrast, from the perspective of GPs, informants described structured feedback on antibiotics as limited by gaps in information, such as the absence of data on how long a patient actually took antibiotics.

### Meso level

Two main types of arrangements are in place for providing feedback at the meso level. These include regional groups, specifically care groups, as geographically defined networks of healthcare providers which provide feedback to affiliated practices. Additionally, research and academic GP networks, such as the Nivel primary care database and GP practices organised around academic hospitals, also conduct research on specific indicators of interest to affiliated GPs.

Dutch professional associations for GPs (eg, National Association of GPs, Dutch GP Association) and pharmacists (eg, Royal Dutch Society for the Promotion of Pharmacy) provide feedback on prescribing for professional development purposes. In the sphere of community pharmacists, the number of medication reviews, participation in pharmacotherapy meetings (FTOs), as well as indicators related to dispensing amounts are regularly measured.

Uses of primary care prescribing data for monitoring purposes by meso level organisations was described to typically include volume indicators related to the total prescriptions annually, compared with previous years and by age groups. Active monitoring of benzodiazepines at the meso level was noted to have decreased following changes in reimbursement coverage from January 2009. As one informant explained:

Around three quarters of prescriptions for benzodiazepines are not reimbursed and data [used] relies on the reimbursement claims. (Association–8)

Moreover, as another informant described with regards to monitoring the uses of prescribing data more locally (eg, by regions), overall activity is currently limited.

The discussion on the use of prescriptions at the moment is taking place at the national-level and at the local level but not at the regional-level. This may and is likely to change in the coming years as care groups are more actively involved in the regional implementation of policies. (Association–15).

## Macro level

At the macro level, pharmacy and claims data are used for strategy development, system performance measurement and quality assurance purposes. Indicators related to the tracer prescriptions are also reported for international comparisons (eg, total volume of antibiotics for systemic use, elderly patients with prescription of long-term benzodiazepines or related drugs and overall volume of opioids prescribed). A number of policy initiatives are in place to monitor antibiotic prescribing and opioids. However, with regards to benzodiazepines, informants described this as a less pertinent priority following the change in reimbursement resulting in an overall decreasing trend in the number of benzodiazepines prescribed.

## Optimising the use of primary care prescribing data

Five main themes were identified as areas for optimising the use of primary care prescribing data: (1) measuring what matters, (2) increasing data linkages, (3) improving data quality, (4) facilitating data sharing and (5) optimising fit for use analysis. Theme 1 pertains to methodological considerations about the indicators in use, while themes 2, 3 and 4 relate to contextual considerations, specifically, the underlying information system and regulations. The last theme is found to reflect managerial considerations influencing an indicator's use in practice. The themes are described to follow.

## Measuring what matters

'We have the data. We don't have the right indicator' (Health professional–2). Similar statements were made in reference to indicators currently in use, in particular at the micro level. Specifically, the absence of indicators to monitor the stop date of prescriptions were noted, despite the relevance of this information to limit over-represcriptions. Information on the stop date for prescriptions was described of growing importance. Notably, as GPs increasingly work in teams and multiple practices, there is greater potential for represcribing to go unnoticed. Similarly, the absence of indicators that distinguish between new versus repeat refills, as well as indicators for monitoring 'de-prescribing' were noted as an information gap, especially for measuring quality of chronic care services.

The lack of indicators to measure the appropriateness of prescriptions was also raised:

Instead of receiving, 'this month you prescribed this many antibiotics' to know 'this month you prescribed this many antibiotics for this many patients diagnosed with infections' can provide more insights into a GP's actual performance. (Association–15)

Dispensing data we have is really useful for the overall consumption, but it is limited to assess the quality of care. For example, for antibiotics use and to determine the appropriateness of the use you really need to have the diagnosis data. (Association–1)

## Increasing data linkages

The interoperability of data systems was a recurrent theme across informants from all levels of the healthcare system. The challenge to link data sources was described both *within* primary care (GPs and community pharmacists) but also *across* levels (GPs, hospitals and community pharmacists). At present, a reliance on manual data exchange between stakeholders was depicted (eg, patients providing data to community pharmacists following hospital discharge, pharmacists providing data to GPs at FTO meetings). While in part a consequence of privacy regulations, informants underscored issues of fragmentation and siloed data systems.

In a perfect world we would have more linkages between the GP databases and that of the pharmacy. Because we know that the systems in the GP practice is lacking some of the information that is available to the pharmacist. Also, what is prescribed in hospital. We need a connection between these systems to create really good indicators. (Association–8)

In the absence of data linkages within primary care as well as specialised care, informants emphasised the implications on the completeness of data and potential to 'see the whole picture' (EHR supplier–10).

## Improving data quality

The quality of coding is a fundamental challenge to the secondary use of prescribing data. As one informant described:

If a GP wants to prescribe antibiotics, then they can also change the code, for example, if someone presents with a possible infection and I see they are quite sick, I can code this differently. (Association–15)

Additionally, the poor quality of coding itself was raised:

In many GP practices at the moment there is simply not enough attention for the quality of the prescription [coding]. GPs are using very old codes [medication codes] in their prescriptions, simply by way of copying their old prescriptions. (EHR supplier–10)

The pertinence of this issue is well-studied (eg,[19]) and is underscored in projects such as Nivel's formulary-oriented

prescribing initiative (Formulariumgericht voorschri-jven),[43] where attention is called to improving the quality of GP prescribing.

### Facilitating data sharing

Informants raised privacy barriers as a key cause for untapped opportunities to stimulate data sharing across the healthcare system. The European General Data Protection Regulation (GDPR) and national privacy and data ownership policies were referenced as challenges to the sharing and connecting of different sources of data. As one informant described: 'It is a political issue of clarifying who is in fact the owner of the data' (Association–14). Informants emphasised the importance of addressing privacy constraints and data sharing in order to allow for more extensive uses.

### Actionable analysis

Informants across all levels described limitations regarding the usefulness of analysed data to inform decision-making. Specifically, at the micro level opportunities to improve the use of comparators were detailed. For example, the current practice of providing an individual GP with feedback on their performance in relation to the national level was described as too aggregate a summary. The consequence, as one informant noted, is a tendency to defer accountability and cite the uniqueness of one's practice population as a cause for deviating trends. In another example, an informant described the compromised actionability of feedback:

Informing 'you are adhering to guidelines in 80% of prescriptions issued' is not helpful to a GP. It leaves unanswered questions, such as, what patients were involved. (Association–8)

Other obstacles described included the ability to discriminate performances to capture practice variation, with one informant stating: 'the problem with the analysis is that the results are not wide. Everyone ends up at the same place' (Insurer–19). Additionally, analysed data fails to capture at-risk patients and vulnerable groups, of relevance across micro-meso-macro contexts. As one informant described from the perspective of pharmacists, current indicators and approaches to analyse information are strained to provide a clear direction for improvement related to care for patients with the greatest needs:

I think we need more data to better target the patients that are in need of additional care. Not everyone needs additional, specialised care. It's the 20% that needs additional, specialized care, and for that, our pharmaceutical database is not sufficient. (Association–1)

Obstacles to analyse data that meet the timeliness needs of decision-makers were also described as a hurdle to the optimal use of data. One informant detailed this challenge extends to the timeliness and accessibility of how data is ultimately delivered to end-users: 'We miss a dashboard or system that would allow gaining access and make use of the available data' (Association–12).

## DISCUSSION

In this study, we set out to investigate sources, secondary uses, and key considerations for optimising primary care prescribing data and its actionability for quality of care related decision-making. Much of the existing literature on measurement for improving primary care prescribing focuses on implementation sciences and practice-level interventions (eg,[44–46]). There is also a dedicated field of research on improving prescribing through interventions in direct patient care (eg,[27–29]). We add to this evidence by adopting a healthcare performance intelligence lens and exploring the secondary uses of primary care prescribing data for learning and improvement in the Dutch healthcare system.

Our study confirms the numerous secondary uses of electronic primary care data across the clinical, organisational and policy context of the healthcare system in the Netherlands. Nonetheless, data are constrained by professional and organisational siloes and perceived privacy constraints that compromise the completeness of information for secondary uses. Importantly, resolving data-related barriers alone will not increase the use of prescribing data. In addition, attention to the development of strategic, purpose-driven indicators and their embedding in systems of governance and managerial cycles, is needed. These findings are further described to follow.

First, with regards data sources, the incompleteness of individual primary care prescribing data sources is a known limitation.[47 48] Our findings regarding challenges to link available data sources are consistent with recent reporting on the Dutch health information system in general[17] and ultimately, common to many European routine healthcare information systems.[48–50] Importantly, while often justified as a legal constraint, regulations like GDPR in fact leave much room for national legislation.[51] Recent Dutch initiatives like the 'Electronic Data Exchange in Health Care Bill'[52] and national quality and information standards for the exchange of medication data[53 54] are important steps being taken for more integrated data at the point of care. However, the same level of policy attention remains needed to ensure that complete data is available for secondary uses.

Second, our findings suggest existing indicators require further development by prescription type and their intended uses. A general fixation on the scientific merits of an indicator in the field of performance measurement has put attention to the development and selection of indicators based on their validity and reliability.[55] However, we observe this focus on scientifically strong indicators in the context of primary care prescribing has distracted from the selection of prescribing indicators based on strategic measurement goals. Our finding that indicators are not differentiated by individual prescription types and

information needs of stakeholders attests to this. Similar to previous studies (eg,[13 56 57]), informants described differences in their desired type of information. The development of indicators with a focus on the *use* and *users* of prescribing indicators to achieve performance goals is needed across the micro-meso-macro level.

Third, putting data to work requires an enabling institutional environment.[58] Realising learning and improvement in practice across the healthcare system is a matter of good governance and management. Challenges to use primary care prescribing data underscores that the use of indicators is a process. The effective use of indicators relies also on governance considerations such as the mandates of stakeholders and alignment of resources.[59] In the absence of an enabling governance system spanning all levels of the healthcare system,[60–62] policy priorities like managing antibiotic resistance and responding to the opioid epidemic, risk to remain solely high-level goals rather than cascading the system. Other governance and managerial considerations include how that information is returned to end-users, such as in reports or dashboards, and ultimately, processes for reflection on the information, need to be fostered and tailored to different stakeholders.

Finally, we note that despite the range of stakeholders and activities found at each level of the healthcare system, we observe that the current uses of prescribing data are primarily for internal, provider-oriented purposes rather than for public reporting and accountability. However, the prescribing data available has a range of potential uses for the public. These uses include for accountability purposes but also for learning regarding side effects and harms related to the inappropriate use of antibiotics or longer-term use of opioids and benzodiazepines, and ultimately, have an important role to play in the patient safety agenda.

### Strengths and limitations

This study was enriched by the diverse engagement of stakeholders across all levels of the Dutch healthcare system, resulting in a thorough qualitative dataset. The advanced digitalisation and secondary uses of primary care data in the Dutch setting may be transferable to other data-rich contexts while also serving as an aspirational example for those at an earlier stage of development. For the purposes of this study and its scope, we focused on the use of indicators for antibiotics, benzodiazepines and opioids and the results, therefore, may not reflect the nuances of all prescription types. Other types of medications, such as for chronic conditions, were excluded as the management of healthcare needs is multifaceted and the appropriate rate of prescriptions is highly patient, disease and risk-factor specific. All interviews took place in English with native Dutch-speakers. Finally, the study by design is exploratory in nature. Therefore, patterns and experiences by stakeholder and data types require testing with a larger sample before they can be generalised. Relatedly, the study has put focus on the secondary uses of prescribing data and, therefore, may not be generalisable to uses for direct patient care, such as in shared decision-making and patient education.

### CONCLUSIONS

Drawing on the expertise of the diverse sample of stakeholders interviewed, we described the information potential of electronic clinical, administrative and claims prescribing data for secondary quality of care-related uses. Informants stressed the unique strengths and limitations of available data sources, with the incompleteness of each individually a key challenge. While primary care prescribing data is in use across the Dutch healthcare system, existing indicators require further development. In the case of antibiotics, this is found as a need to better indicate the appropriateness of prescriptions and for benzodiazepines and opioids, to monitoring their long-term use. Beyond methodological considerations about the indicators themselves, contextual considerations related to the information system and regulations as well as managerial considerations influencing an indicator's use in practice are areas identified for further prioritisation. To curb societal concerns like antibiotic resistance and the misuse of opioids and benzodiazepines, the availability of prescribing data alone is insufficient. Available data sources must be linked and made actionable through fit for purpose and fit for use indicators applied at all levels of the healthcare system.

**Acknowledgements** We thank all participants who provided their time and expertise to this study as well as the time of the Data Expert Community members and reviews by HealthPros fellows, Karin Hek and other Nivel colleagues. We also thank the reviewers for their thoughtful and constructive feedback.

**Contributors** EB, RAV, LR, NK and DK conceptualised the study. EB with the support of RAV and LR conducted data collection. EB prepared the manuscript. All authors provided feedback and contributed to revising the manuscript. All authors approved the final version. The guarantor (EB) accepts full responsibility for the finished work and/or the conduct of the study, had access to the data, and controlled the decision to publish.

**Funding** This work was carried out by the Marie Skłodowska-Curie Innovative Training Network for Healthcare Performance Intelligence Professionals (HealthPros) that has received funding from the European Union's Horizon 2020 research and innovation programme under grant agreement Nr. 765141.

**Competing interests** None declared.

**Patient and public involvement** Patients and/or the public were involved in the design, or conduct, or reporting, or dissemination plans of this research. Refer to the Methods section for further details.

**Patient consent for publication** Not applicable.

**Ethics approval** The research protocol was developed in accordance with the ethical requirements of the primary research affiliation to Amsterdam University Medical Centers of the University of Amsterdam and relevant Dutch ethics guidelines. The research adheres to the Dutch ethics guidelines stated in the 'Medical Research Act with People (Wet medisch-wetenschappelijk onderzoek met mensen (WMO) (Dutch), in BWBR0009408, W.a.S. Ministry of Health, Editor. 1998: Hague, Netherlands',51) for which exception applies as no human data were retained and voluntary informed consent of participants was deemed adequate by the authors. Participants gave informed consent to participate in the study before taking part. To ensure informed voluntary participation, informants contributing to this study provided written informed consent to participate during the recruitment stage and restated their consent verbally at the start of interviews. All interview

data has been anonymised. Confidentiality was assured by referring to informants by stakeholder type and an assigned number (eg, Health professional–1).

**Provenance and peer review** Not commissioned; externally peer reviewed.

**Data availability statement** All data relevant to the study are included in the article or uploaded as supplementary information. The dataset supporting the conclusions of this study are included within the article and its supplementary files.

**ORCID iD**
Erica Barbazza http://orcid.org/0000-0001-7621-1638

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
