## [Reviewer comments · BMJ Open]

ARTICLE DETAILS

TITLE (PROVISIONAL)	Optimising the secondary use of primary care prescribing data to improve quality of care: a qualitative analysis
AUTHORS	Barbazza, Erica; Verheij, Robert A; Ramerman, Lotte; Klazinga, Niek; Kringos, D

VERSION 1 – REVIEW

REVIEWER	Pristas, Ivan Croatian Institute of Public Health, Health Informatics and Biostatistics
REVIEW RETURNED	01-Apr-2022

GENERAL COMMENTS	Dear authors, This is a great example of qualitative comprehensive approach in identifying use of health information for continuous improving processes and quality. I personally enjoyed reading it. Now to the minor remarks: 1. Although number of stakeholders involved looks almost complete, it should be more clarified. Namely, sampling involved more than 20 stakeholders, and 26 stakeholders were involved through 28 interviewees. 25 contacted persons dropped off. It is unclear did they belong to other stakeholders and how many stakeholders were omitted in the study in total. If any, how important and what is the potential influence on findings (results and conclusions).2. The article provides clear and strong evidence in the results part, but misses the opportunity to use it strongly enough towards the readers in the discussion part. The discussion part is rather smooth and has too distanced academic approach. There is no need for this. I would advise to change style, try to rephrase the discussion part where possible, to omit redundant phrases from the previous chapters, and to go straight to the point in order to maintain clear messages for all potential readers on micro, meso and macro levels. We are all aware that there are big hopes, investments and initiatives for improving use of health data, analyses and information. However, the use of those is still significantly varying depending on the user's positions, mandates, fitness for purpose and use, contexts, quality, understanding and other incentives including willingness for action. That maintains the huge gap between data availability and data use.
---

	Therefore, measures for improving health information use should be somehow classified and prioritized, hopefully by the informed and experienced authors' proposal in the discussion part. Besides the minor suggestions, great work is being done in this inspirational research!
--	---

REVIEWER	Okoli, Grace Queen Mary University of London, Centre for Primary Care and Mental Health
REVIEW RETURNED	25-Apr-2022

GENERAL COMMENTS	Reviewer- Grace N Okoli (https://orcid.org/0000-0002-1740-4058) Q3. Is the study design appropriate to answer the research question? Lacks patient and public stakeholder involvement. Q9. Do the results address the research question or objective? The study describes limitations in the use of primary care prescribing datasets from the perspective of the chosen stakeholder but does not address how it can be optimised in a patient care. It highlights the limitations rather than optimisation. Table is a good description of the benefits and limitation of primary care prescribing datasets TITLE Optimizing the secondary use of primary care prescribing data to improve quality of care: a qualitative analysis. Q10. Are they presented clearly? There are issues with sentence structure and clarity in key areas of the manuscript.. Q12. Are the study limitations discussed adequately? No as outlined in the comments on the manuscript, it lacks patient and public feedback and this is not addressed adequately overall the representation from different stakeholder discussed as the clinical perspective is represented by only one person. Q15. Is the standard of written English acceptable for publication? There is a need f for improvement in the clarity of the manuscript particularly when it makes key points
---

VERSION 1 – AUTHOR RESPONSE

Reviewer 1

Dear authors, this is a great example of qualitative comprehensive approach in identifying use of health information for continuous improving processes and quality. I personally enjoyed reading it.

Now to the minor remarks:

Although number of stakeholders involved looks almost complete, it should be more clarified. Namely, sampling involved more than 20 stakeholders, and 26 stakeholders were involved through 28 interviewees. 25 contacted persons dropped off. It is unclear did they belong to other stakeholders and how many stakeholders were omitted in the study in total. If any, how important and what is the potential influence on findings (results and conclusions).

Thank you for this comment and the opportunity to clarify the distribution of non-participants. In revising, we have provided a more detailed breakdown of non-respondents to specify this not only by “type”
span style="font-family:'Times New Roman'; font-weight:bold"> but also by the distribution across different target informants. This has been added below and in the supplementary files of the manuscript. The breakdown shows that non-respondents ranged all levels of the target

informant types and that none of the level/groups were overly non-responsive. We also took the opportunity to better classify non-participants as “no reply” (N=12) and “unavailable” (N=3). The category, previously labelled as “suggested alternate” (N=10), have been relabelled as “contact mediating.” We view this term to more accurately reflect their involvement. That is, their thoughtful consideration of the request and suggestion of a colleague within their team or organization best suited for participating. This snowballing of recommendations was part of the sampling strategy. In text, in the Result section “Characteristics of informants” these details on non-participants and contact mediating informants have been added, together with reference to the new table in Supplementary file 3.

New Table S3.1. Elaborated breakdown of informants and non-participants

Characteristics	Total informants N=28		Non-participants N=25		
	n	%	No reply	Unavailable	Contact mediating
Healthcare system level (context)					
Micro (clinical)	1 (4)	4	1	2	1
Meso (organizational)	11	39	6	0	5
Macro (policy)	9	32	3	0	2
Cross-cutting (research, EHR supplier)	7	25	2	1	2
Type of stakeholder					
Association (patient, professional)	8	29	3	0	2
Care group (network)	2	7	0	1	0
Government health agency	9	32	3	0	2
Health professional	1 (4)	4	0	1	1
EHR supplier	4	14	1	0	2
Insurer	1	4	3	0	3
Research	3	11	2	1	0
Gender					
Female	8	29	4	2	4
Male	20	71	8	1	6

EHR: Electronic health record.

^aNumbers in round brackets indicate the total number of informants when individuals with multiple affiliations are accounted for.

The article provides clear and strong evidence in the results part, but misses the opportunity to use it strongly enough towards the readers in the discussion part. The discussion part is rather smooth and has too distanced academic approach. There is no need for this.

I would advise to change style, try to rephrase the discussion part where possible, to omit redundant phrases from the previous chapters, and to go straight to the point in order to maintain clear messages for all potential readers on micro, meso and macro levels.

We are all aware that there are big hopes, investments and initiatives for improving use of health data, analyses and information. However, the use of those is still significantly varying depending on the user's positions, mandates, fitness for purpose and use, contexts, quality, understanding and other incentives including willingness for action. That maintains the huge gap between data availability and data use.

Therefore, measures for improving health information use should be somehow classified and

prioritized, hopefully by the informed and experienced authors' proposal in the discussion part. Besides the minor suggestions, great work is being done in this inspirational research!

Thank you for this important reflection and opportunity to improve the discussion section of the manuscript. Reflecting on the points made above, we conducted a major revision of the discussion. In revising, we have been attentive to avoid repeated statements from the findings. We have also put emphasis on calling clearer attention to embedding the use of prescribing data in governance and managerial cycles. We argue in agreement with the points noted above to emphasize that the barriers to optimizing the use of prescribing data sources are a symptom of the institutional environment. In doing so, we have put focus to the importance of clear strategic goals to guide the uses of prescribing indicators, and the role of good governance and management to ensure aspects related to the processes of using indicators are in place (eg, mandates, resources).

Reviewer 2

Is the study design appropriate to answer the research question? Lacks patient and public stakeholder involvement.

Thank you for the opportunity to clarify this point. Directly exploring the uses of primary care prescribing data with patients was outside the scope of the study. Focus was put to the secondary uses of primary care prescribing data, rather than direct patient care. Importantly, patients as a stakeholder involved in the secondary use of primary care prescribing data was captured through the sampling of patient associations, with a representative from the Dutch Patient Federation participating as an informant. We view their involvement as a proxy for exploring the secondary uses of primary care prescribing data from the perspective of patients.

We have taken a number of steps to better clarify the involvement of patients and its implications on how the results are interpreted. Firstly, we have added to the introduction an explicit statement that the perspective of patients and uses of prescribing data for shared decision-making and patient care was outside the scope of the study. Second, we have made more explicit that sampling included the perspective of patient associations as this was not previously stated and only included in the reporting on types of stakeholders. Third, in the discussion section, we have added a reflection on indirect observation that the uses identified are mainly for internal improvement and uses by patients could be strengthened.

Do the results address the research question or objective? The study describes limitations in the use of primary care prescribing datasets from the perspective of the chosen stakeholder but does not address how it can be optimised in a patient care. It highlights the limitations rather than optimisation. Table is a good description of the benefits and limitation of primary care prescribing datasets TITLE Optimizing the secondary use of primary care prescribing data to improve quality of care: a qualitative analysis.

Thank you for noting this point. We took the opportunity of revising to more explicitly address the scope of the study's focus on secondary uses of prescribing data rather than on direct patient care. This has been revised in the introduction (see highlighted text in the main document). We included examples of literature that focus specifically on the patient perspective in prescribing. Nonetheless, we view the optimisation of the secondary uses of primary care prescribing data as an intermediary step to improving care. In addition to the other changes noted above in this regard, we have also added to the discussion section the results may not be generalizable to other uses beyond those explicitly studied, like patient care.

Are they presented clearly? There are issues with sentence structure and clarity in key areas of the manuscript.

Thank you for calling our attention to this. In reviewing, the manuscript has been proofread with the aim to improve sentence structure and clarity throughout. This editing has intended simplify the language and make key messages clearer. Editorial changes have been made throughout the manuscript silently.

Q12. Are the study limitations discussed adequately? No as outlined in the comments on the manuscript, it lacks patient and public feedback and this is not addressed adequately overall the representation from different stakeholder discussed as the clinical perspective is represented by only one person.

As noted above, we have added to the limitations an explicit statement regarding the generalizability of the findings to patient care. Importantly, we do take the opportunity to call attention to the sample. In the results, we note: “three additional informants were both health professionals and affiliated to another stakeholder.” In revising, we have incorporated this into the table for ease of reading, adding: “as signalled by totals included in round brackets in Table 1.” We found this range of perspectives sufficient and balanced with the diversity of other types of stakeholders sampled.

Q15. Is the standard of written English acceptable for publication? There is a need for improvement in the clarity of the manuscript particularly when it makes key points

We trust this comment has been resolved through the exercise of proofreading completed and the revisions to the discussion to address the comment of Reviewer 1 that introduced changes to improve the clarity of messages.

Page 1 of 3